# Selective Oxidation of Glycerol via Acceptorless Dehydrogenation Driven by Ir(I)-NHC Catalysts

**DOI:** 10.3390/molecules27227666

**Published:** 2022-11-08

**Authors:** M. Victoria Jiménez, Ana I. Ojeda-Amador, Raquel Puerta-Oteo, Joaquín Martínez-Sal, Vincenzo Passarelli, Jesús J. Pérez-Torrente

**Affiliations:** Departamento de Química Inorgánica, Instituto de Síntesis Química y Catálisis Homogénea-ISQCH, Universidad de Zaragoza-C.S.I.C., 50009 Zaragoza, Spain

**Keywords:** glycerol, lactic acid, dehydrogenation, hydrogen release, N-heterocyclic carbenes, iridium, homogeneous catalysis

## Abstract

Iridium(I) compounds featuring bridge-functionalized bis-NHC ligands (NHC = N-heterocyclic carbene), [Ir(cod)(bis-NHC)] and [Ir(CO)_2_(bis-NHC)], have been prepared from the appropriate carboxylate- or hydroxy-functionalized bis-imidazolium salts. The related complexes [Ir(cod)(NHC)_2_]^+^ and [IrCl(cod)(NHC)(cod)] have been synthesized from a 3-hydroxypropyl functionalized imidazolium salt. These complexes have been shown to be robust catalysts in the oxidative dehydrogenation of glycerol to lactate (LA) with dihydrogen release. High activity and selectivity to LA were achieved in an open system under low catalyst loadings using KOH as a base. The hydroxy-functionalized bis-NHC catalysts are much more active than both the carboxylate-functionalized ones and the unbridged bis-NHC iridium(I) catalyst with hydroxyalkyl-functionalized NHC ligands. In general, carbonyl complexes are more active than the related 1,5-cyclooctadiene ones. The catalyst [Ir(CO)_2_{(MeImCH_2_)_2_CHOH}]Br exhibits the highest productivity affording TONs to LA up to 15,000 at very low catalyst loadings.

## 1. Introduction

Currently, the mainstream of commodities is produced from non-renewable fossil fuel resources. However, the non-sustainable nature of fossil resources demands the development of cleaner methodologies to produce chemicals from alternative resources [1]. In this regard, the use of biomass in the biodiesel industry has led to a vast worldwide production of glycerol that is available on a large scale. In fact, the main byproduct in biodiesel processing is the large quantities of crude glycerol, which represent approximately 10 wt% of the total biodiesel [2]. Lactic acid (LA) is widely used in the food, cosmetics, pharmaceuticals, polymers and fine chemicals industries [3]. The increasing demand for this chemical platform and the disadvantages of the current production of LA from sugar fermentation call for the development of novel methodologies [4]. Therefore, new approaches for the valorization of biomass waste products into value-added chemicals are highly desirable. To date, an attractive method is the catalytic oxidation of glycerol to produce lactic acid (LA) and dihydrogen [5]. In this regard, the search for efficient and selective catalysts should have a favorable impact on the development of this methodology. 

Heterogeneous catalysts have already proven successful in the dehydrogenation of glycerol to LA. However, this methodology presents some weaknesses as harsh reaction conditions and low selectivities [6,7,8]. Recently, homogeneous catalysts based on Ir, Ru or Fe for the conversion of glycerol to LA under mild conditions have been developed [9,10,11]. In this regard, transition metal catalysts based on PNP pincer ligands have shown remarkable activity in the dehydrogenation of a range of substrates, such as formic acid [12], primary and secondary alcohols [13], including methanol [14,15], nitrogen-containing heterocycles [16] or even oxidative dehydrogenation of glycerol to lactate [17,18]. Currently, Ir-NHCs complexes play a prominent role in the area, as they have been revealed as robust catalysts for a number of dehydrogenation reactions, including the selective oxidation of glycerol [19]. In particular, carbonyl Ir-NHC catalysts such as [Ir(CO)_2_(IMe)_2_]^+^ (IMe = 1,3-dimethylimidazol-2-ylidene) [20], [Ir(CO)_2_{MeIm(pyridin-2-ylmethyl)}]^+^ [21] or even a related water-soluble zwitterionic iridium catalyst featuring NHC ligands with sulfonate-functionalized wingtips [22,23] have shown an excellent catalytic performance in the selective acceptorless dehydrogenation of glycerol to lactate. Furthermore, it has been shown that bis-NHC ligands provide additional stability to the organometallic complexes due to the chelate coordination mode that makes them more resistant to harsh reaction conditions [24]. In fact, recently, mono- and bimetallic iridium complexes supported by triscarbene ligands were applied to the dehydrogenation of biomass-derived glycerol affording hydrogen and lactate with outstanding turnover frequencies [25]. In addition, the coordination polymer derived from a rigid bis-benzimidazolium precursor featuring Ir(CO)_2_ metal fragments assembled via a self-supported strategy allows its recycling more than 30 times without any loss of activity [26]. The excellent catalytic performance exhibited by the bimetallic catalytic system [25], the coordination polymer [26] and the NHC-pyridine iridium catalyst [21] is likely a consequence of the effective suppression of inactive binuclear iridium species formed under the catalytic reaction conditions.

In this context, we envisaged the potential of bridge-functionalized bis-NHC ligands containing potentially hemilabile carboxylate or hydroxyl groups as more stable metal-ligand platforms with improved solubility in polar media. Therefore, the present work focuses on the synthesis of a series of bridge-functionalized bis-NHC iridium(I) complexes with the goal of assessing their catalytic performance for the acceptorless dehydrogenation of glycerol to lactic acid. With this aim, we have selected the bis-imidazolium salts **A** and **B**, precursors of the linker-functionalized bis-NHC ligands 1,1-bis(*N*-methylimidazol-ylidene)-acetate and 1,1′-(2-hydroxy-1,3-propanediyl)bis [3-methyl-imidazol-ylidene), respectively. Imidazolium salt **A** [27] has provided access to a stable metal-ligand platform for the design of rhodium and iridium complexes with application in a range of catalytic transformations [28,29,30,31], whereas imidazolium salt **B** has been successfully applied for the preparation of palladium, copper, silver and gold complexes with catalytic applications [32,33,34]. In addition, related iridium(I) complexes derived from the bridge unfunctionalized bis-imidazolium salt, 1,1′-methylene-bis(3-methylimidizolium) (**C**) [35], and 1-(3-hydroxypropyl)-3-methyl-imidazolium) (**D**) [36,37] have also been prepared for comparative purposes ([Fig molecules-27-07666-ch001]).

## 2. Results and Discussion

### 2.1. Synthesis and Characterization of bis-NHC Iridium(I) Complexes

A series of mononuclear bis-NHC iridium(I) catalysts have been synthesized by reaction of the carboxylate- and hydroxy-functionalized bis-imidazolium salts, **A** and **B**, respectively, with appropriate iridium-diene precursors. In addition, a related iridium(I) compound with the unfunctionalized bis-imidazolium salt **C** has also been prepared ([Fig molecules-27-07666-ch001]). Interestingly, a number of carbonyl rhodium and iridium complexes have found application in several catalytic and thus, related carbonyl bis-NHC iridium(I) have also been prepared by carbonylation of the diene precursors.

#### 2.1.1. Carbonylation of [Ir(cod){(MeIm)_2_CHCOO}]

We have recently reported the synthesis of the zwitterionic iridium(I) [Ir(cod){(MeIm)_2_CHCOO}] (**1**) (cod = 1,5-cyclooctadiene; MeIm = 3-methylimidazol-2-yliden-1-yl) as a catalyst precursor for the hydrogenation of CO_2_ to formate in water [30]. The synthesis of **1** was accomplished by in situ deprotonation of **A** upon reaction with [Ir(μ-OMe)(cod)]_2_ and NaH in MeOH. Bubbling of carbon monoxide through a red suspension of **1** in CH_2_Cl_2_ for 10 min at room temperature afforded a pale yellow solution from which the five-coordinated iridium(I) complex [Ir(CO)(cod){(MeIm)_2_CHCOO}] (**2**) was isolated in 78% yield (Figure 1). The spectroscopic data for **2** agree with the solid-state structure determined by X-ray diffraction methods (see below). The ^1^H NMR spectra of **2** showed a set of characteristic resonances indicative of the presence of a coordinated cod ligand. The single absorption band at 1944 cm^−1^ in the ATR-IR spectrum was assigned to the carbonyl ligand, which was observed as a singlet at δ 179.0 ppm in the ^13^C{^1^H} NMR spectrum. Furthermore, the ESI+ spectra showed a peak at *m*/*z* 541.1 with the right isotopic distribution corresponding to the protonated molecular ion. Interestingly, the =CH protons of the cod ligand showed a broad resonance at δ 3.90 ppm in the ^1^H NMR and a singlet at 71.3 ppm in the ^13^C{^1^H} NMR spectrum, which suggests a dynamic behavior in solution typical of pentacoordinated complexes [38].

Carbonylation of **1** for longer reaction times easily produces the substitution of the diene ligand to afford the square-planar complex [Ir(CO)_2_{(MeIm)_2_CHCOO}] (**3**). Thus, stirring of a red suspension of **1** in CH_2_Cl_2_ under a carbon monoxide atmosphere for 16 h gave a yellow solution from which **3** was isolated as a yellow solid in 55% yield. This result suggests that the five-coordinated compound **2** is an intermediate in the formation of the square planar carbonyl complex **3** by carbonylation of **1** (Figure 1). The ESI+ spectra of **3** showed a peak at *m*/*z* 541.1 with the right isotopic distribution corresponding to the protonated molecular ion. In addition, the ATR-IR spectrum showed two strong carbonyl ν(CO) bands at 2067 and 2007 cm^−1^, in accordance with a *cis* dicarbonyl complex. In contrast to the parent diene complex **1**, the ^1^H NMR of **3** in CD_3_OD showed the presence of a single isomer. In addition, the CHCOO resonance was found rather downfield shifted, around δ 6.7 ppm, which could be a consequence of the presence of stronger π-acceptor ligands in the coordination sphere of the metal center. On the other hand, the equivalent carbonyl ligands were observed as a singlet at δ 183.2 ppm in the ^13^C{^1^H} NMR spectrum.

#### 2.1.2. Synthesis and Reactivity of [Ir(cod){(MeImCH_2_)_2_CHOH}]Br

To study the possible influence on the catalytic activity of the bridge-chain length, a related hydroxy-functionalized bis-NHC ligand with three carbon atoms in the bridge between both imidazole-2-carbene moieties was selected. The hydroxy-functionalized bis-imidazolium salts [{(RImH)CH_2_}_2_CHOH]Br_2_ (R = Me, ^i^Pr, Bn) and their rhodium(I) complexes were synthesized by Herrmann et al. [39]. We have prepared the related iridium(I) compound [Ir(cod){(MeImCH_2_)_2_CHOH}]Br (**4**) following a two-steps procedure similar to that described for the synthesis of rhodium(I) complexes and related unbridged bis-NHC iridium(I) compounds synthesized by us [40] (Figure 2). First, a solution of the ethoxo-bridge iridium dimer [Ir(µ-OEt)(cod)]_2_ was prepared by reaction of [Ir(µ-Cl)(cod)]_2_ with an excess of NaH in ethanol. Then, the solution was added to a solution of [{(MeImH)CH_2_}_2_CHOH]Br_2_ in ethanol to give an orange solution from which **4** was isolated as a microcrystalline shiny orange solid in 80% yield. In this synthetic methodology, the bridging ethoxo groups in the iridium dimer are responsible for the deprotonation of one imidazolium moiety, whereas the deprotonation of the second one takes place due to the excess of NaOEt in the reaction medium.

The ESI+ mass spectra of **4** showed a peak at *m*/*z* 521.0, corresponding to the molecular ion with the right isotopic pattern. However, in contrast to the rhodium complex, which exists as a single isomer in solution [39], the ^1^H NMR spectrum of **4** in CD_3_OD or CD_2_Cl_2_ showed duplicate signals for all proton resonances, confirming the presence of two isomers. As can be seen in the ^1^H NMR spectrum in CD_2_Cl_2_ at room temperature, both isomers are in a 2/1 ratio (Figure 1). The imidazole-2-carbene ring protons appeared as two sets of doublets at δ 7.37 and 6.80 ppm (*J*_H-H_ = 1.9 Hz) for the major isomer and at 7.23 and 6.88 ppm (*J*_H-H_ = 1.9 Hz) for the minor isomer, whereas a broad singlet at δ 6.25 ppm was observed for the hydroxy proton of both isomers. The equivalent N-Me protons of both isomers were observed as two singlets at δ 3.81 ppm for the minor isomer and 3.80 ppm for the major isomer. Two broad signals at δ 4.14 and 4.09 ppm include the =CH olefin protons of both isomers, whereas the >CH_2_ protons appeared as a set of broad resonances in the range 2.43–1.95 ppm. Nevertheless, the olefin resonances for the cod ligand were observed as two sets of two singlets at δ 77.2, 76.0 ppm (major isomer) and 77.9, 76.2 ppm (minor isomer) in the ^13^C{^1^H} NMR spectrum, which agrees with the presence of a symmetry plane in each one of the isomers.

Most likely, both isomers result from the different disposition of the hydroxy substituent on the eight-membered metallacycle with a boat-chair conformation, namely pseudo-axial and pseudo-equatorial positions. The major isomer should correspond to the one having the hydroxy group in the pseudo-axial position, similar to the solid-state structure (see below). Interestingly, both isomers can be easily interconverted by inversion of the metallacycle followed by twisting. However, the ^1^H-^1^H-NOESY spectrum shows no exchange cross-peaks between the main resonances of both isomers, pointing to a high-energy barrier process, and consequently, both isomers do not interconvert in solution. Finally, it is worth mentioning that the substituent in axial position at carbon-5 in a hypothetical boat-boat conformation of the eight-membered metallacycle remains close to the metal center enabling for anagostic interactions, which is not possible in the boat-chair conformation [41].

Carbonylation of [Ir(cod){(MeImCH_2_)_2_CHOH}]Br (**4**) in CH_2_Cl_2_ for 10 min gave a yellow solution of the carbonyl complex [Ir(CO)_2_{(MeImCH_2_)_2_CHOH}]Br (**5**) which was isolated as a microcrystalline yellow solid in 80% yield (Figure 2). The ESI+ mass spectra of **5** showed a peak at *m*/*z* 469.0 with the right isotopic distribution for the molecular ion. In addition, the ATR-IR spectrum displayed two strong absorption bands at 2058 and 1985 cm^−1^ suggesting the existence of two metal-bounded carbonyl ligands in *cis* disposition. As shown by the ^1^H NMR spectrum, solutions of complex **5** in CD_2_Cl_2_ contain two isomers in a 4/1 ratio (see the Appendix A). The imidazole-2-carbene ring protons appeared as two doublets for each isomer in the range δ 7.54–6.93 ppm (*J*_H-H_ = 1.9 Hz), whereas the linker >CH_2_ protons appeared as a set of complex resonances in the range 4.69–4.41 ppm. The equivalent carbonyl ligands of both isomers of **5** were observed as a singlet at δ 182.3 ppm in the ^13^C{^1^H}-apt spectrum. Although duplicate signals for each carbon resonance were observed in the spectrum, a single resonance at δ 169.3 ppm, highfield shifted compared to that of the parent complex **4**, was observed for both isomers.

As described for complex **4**, the two isomers likely arise from the different disposition of the hydroxy substituent in pseudo-axial and pseudo-equatorial positions in the eight-membered metallacycle adopting a boat-chair conformation, with the major isomer having the hydroxy group in pseudo-axial position (see below). As in the case of **4**, the ^1^H-^1^H-NOESY spectrum evidences that both isomers are not interconverting in solution. Although the isomer ratio found in both compounds, 2/1 for **4** and 4/1 for **5**, could be a consequence of the different solubility, an isomerization along the carbonylation process cannot be excluded.

#### 2.1.3. Synthesis of [Ir(cod){(MeIm)_2_CH_2_}]I

The cationic complex [Ir(cod){(MeIm)_2_CH_2_}]I (**6**) has been prepared following the two-step procedure described by Herrmann et al. for the rhodium complex [Rh(cod){(MeIm)_2_CH_2_}]I [42,43]. To the best of our knowledge, the iridium complex **6** has not been reported yet. The reaction of the unfunctionalized bis-imidazolium salt [(MeIm)_2_CH_2_]I_2_ (**C**) with the ethoxo-bridged iridium dimer [Ir(µ-OEt)(cod)]_2_, prepared in situ by reaction of [Ir(µ-Cl)(cod)]_2_ with an excess of NaH in ethanol, in the presence of the excess of NaOEt afforded complex **6** as an orange solid in 63% yield (Figure 3). The ^1^H NMR of **6** in CD_2_Cl_2_ showed two doublets at δ 6.03 and 5.82 ppm (*J*_H-H_ = 11.9 Hz) for the diastereotopic protons of the bridging methylene protons and a broad multiplet at 3.82 ppm for the =CH protons of the cod ligand which agrees with the proposed square planar structure.

#### 2.1.4. Crystal and Molecular Structure of bis-NHC Iridium(I) Compounds

The crystal structure of [Ir(CO)(cod){(MeIm)_2_CHCOO}]·2 CH_3_OH (**2**·2 CH_3_OH) exhibits an asymmetric unit with two crystallographically independent complexes along with four CH_3_OH molecules. Despite the fact that one of the two metal complexes contains a partially disordered cod ligand, the two metal complexes of the asymmetric unit are chemically equivalent and virtually superimposable. On this ground, only the molecular structures of one of them will be discussed herein in detail. The ORTEP view of the solid-state structures of **2**·2 CH_3_OH is shown in Figure 2a, and selected bond lengths and angles are given in Table 1. A distorted trigonal bipyramidal coordination of the metal center is observed at the metal center, with the carbon monoxide ligand in the equatorial plane. Both the bis-NHC ligand, (MeIm)_2_CHCOO^–^, and cod ligands are bidentate, each occupying an equatorial coordination site and an axial one, the bite angles of cod [CT1–Ir–CT2 83.559(8)°] and (MeIm)_2_CHCOO^–^ [C(1)–Ir(1)–C(9) 83.6(2)°] nicely fitting in with the observed coordination geometry. As for (MeIm)_2_CHCOO^–^, the bidentate coordination renders a six-membered Ir1–C1–N5–C7–N8–C9 metallacyle exhibiting a boat conformation, with Ir1 and C7 occupying out-of-plane positions (interplane angles: 34.3°, C7; 26.0°, Ir), with the equatorial Ir1–C9 bond length [2.089(6) Å] slightly longer than the axial Ir1-C1 [2.023(6) Å]. On the other hand, when dealing with the coordinated olefin groups of cod, the axial Ir–CT1 distance [Ir1–CT1 2.1679(2) Å] is significantly longer than the equatorial Ir–CT2 one [Ir1–CT2 2.0183(2) Å], reasonably as a consequence of the trans influence of the NHC occupying the axial position trans to C17–C18. Accordingly, the C17–C18 bond length [1.373(8) Å] is shorter than the C21–C22 one [1.446(8) Å] suggesting a higher metal-ligand back-donation in the case of C21–C22. When dealing with peripheral carboxylato group, similar to what previously observed in related rhodium(I) [27] and iridium(I) [30] [M(diene){(MeIm)_2_CHCOO}] complexes (diene = cod, nbd), the carboxylate group occupies the flagpole position and the carbon-oxygen bond lengths are similar, indicating that the formal negative charge is delocalized. In addition, both oxygen atoms are involved in an O···HO hydrogen bond with lattice methanol (Figure 2a).

Tiny single crystals of **4**·1.75 CH_2_Cl_2_ were grown by slow diffusion of diethyl ether in a solution of **4** in dichloromethane. Weak diffraction patterns were observed (see Experimental Section), and, as a consequence, the quality of the collected data is poor, and the accuracy of the resulting structural determination is low. Nonetheless, the quality of the refined model is good enough to confirm the molecular structure proposed for **4** and will be discussed herein. The asymmetric unit contains two crystallographically independent cations [Ir(cod){(MeImCH_2_)_2_CHOH}]^+^ along with two bromide ions and 3.5 molecules of CH_2_Cl_2_. Since the independent cations [Ir(cod){(MeImCH_2_)_2_CHOH}]^+^ are chemically equivalent, and their structures are virtually superimposable, for the sake of brevity, only one of them will be discussed in the following (Figure 2b). A square planar geometry at the metal center is observed with the bis-NHC ligand occupying two mutually cis positions [C(1)–Ir(1) 2.025(13) Å, C(11)–Ir(1) 2.011(12) Å, C(11)–Ir(1)–C(1) 84.1(5)°] rendering an eight-membered metallacycle that exhibits a boat-chair conformation with the hydroxy group in the pseudo-axial position. Coordinated cod also acts as a bidentate chelating ligand [CT01–Ir1–CT02 86.476(18)°] completing the coordination sphere of the metal center. As for the coordinated olefinic moieties, similar carbon-carbon bond lengths [C(17)–C(18) 1.352(19) Å, C(21)–C(22) 1.391(18) Å] and similar metal-centroid distances are observed [Ir1–CT01 2.0378(5) Å, Ir1–CT02 2.0708(5) Å]. Interestingly, an OH···Br hydrogen bond was observed between the peripheral group O16-H16 of the bis-NHC ligand and the bromide counterion Br1 (Figure 2b). In addition, the NHC cores lie almost perpendicular to the coordination plane (*av*. 83°), and their pitch and yaw angles (see Experimental Section) indicate a distorted arrangement with respect to the corresponding metal-carbon bond. For the sake of comparison, the structure of the related rhodium complex [Rh(cod){(EtImCH_2_)_2_CHOH}]^+^ [39] is similar to that described herein for **4**.

The crystal structure of **5**·CH_2_Cl_2_ contains two crystallographically independent complexes along with two CH_2_Cl_2_ molecules. Since the two cations are chemically equivalent and virtually superimposable, only one of them will be discussed in detail. A distorted square planar coordination geometry is observed (Figure 2c) with the bis-NHC ligand occupying two mutually cis positions [C(1)–Ir(1) 2.056(14) Å, C(11)–Ir(1) 2.083(13) Å, C(1)–Ir(1)–C(11) 80.4(5)°]. Two carbon monoxide ligands complete the coordination sphere [C(17)–Ir(1)–C(1) 171.1(5)°, C(19)–Ir(1)–C(11) 174.0(6)°], similar metal-carbon [C(17)–Ir(1) 1.876(14) Å, C(19)–Ir(1) 1.883(15)°] and carbon-oxygen bond lengths being observed [C(17)–O(18) 1.131(16) Å, C(19)–O(20) 1.151(17)°]. As observed for **4**, the eight-membered metallacycle in **5** adopts a boat-chair conformation with the hydroxy group in the pseudo-axial position. In addition, OH···Br hydrogen bonds are observed between the peripheral O16-H16 group of the bis-NHC ligand and bromide (Figure 2c). Both NHC cores lie almost perpendicular to the coordination plane (81.6, 84.5°), and their pitch and yaw angles indicate that they deviate from the ideal coordination to the metal center, which is reasonably a consequence of crystal packing.

### 2.2. Synthesis and Characterization of NHC Iridium(I) Complexes

The synthesis of a cationic unbridged bis-NHC iridium(I) complex bearing 3-hydroxypropyl functionalized NHC ligands has been carried out to evaluate the flexibility imparted by two monodentate NHC ligands on the catalytic activity. In addition, neutral complexes having a monodentate NHC ligand have also been prepared for comparative purposes.

Compound [IrCl(cod){MeIm(CH_2_)_3_OH}] (**7**) was prepared by deprotonation of the functionalized imidazolium salt [MeImH(CH_2_)_3_OH]Cl (**D**) by the bridging methoxo ligands of compound [{Ir(μ-OMe)(cod)}_2_], and isolated as a yellowish solid in 86% yield. The MS spectrum and the conductivity measurements in acetone support the proposed square planar structure featuring a chlorido ligand and an uncoordinated 3-hydroxypropyl wingtip at the NHC ligand. The formation of the Ir–NHC bond was confirmed by the absence of the ^1^H signal of the C2H group of **D** and the presence of a ^13^C{^1^H} singlet resonance at δ 180.2 ppm assigned to the carbenic carbon atom. The =CH protons of the cod ligand showed four distinct resonances both the ^1^H and ^13^C{^1^H} NMR spectra, thereby suggesting restricted rotation about the Ir–CNHC bond. As a consequence, the methylene protons of the 3-hydroxypropyl wingtip are diastereotopic and showed two well-separated resonances in the ^1^H NMR spectra. On the other hand, the hydroxyl group has been identified indirectly at δ 3.14 ppm since the signal does not correlate with any carbon in the two-dimensional ^1^H,^13^C-HSQC spectra (see Appendix A).

The hydroxy group at the wingtip of compound **7** can be easily deprotonated to afford the neutral alkoxo compound [Ir(cod){κ^2^*C*,*O*-{MeIm(CH_2_)_3_O}] (**8**) with the functionalized NHC ligand exhibiting a κ^2^-C,O coordination mode rendering a seven-membered metallacycle. Thus, the reaction of **7** with one equiv of NaH in anhydrous tetrahydrofuran gave, after removal of the formed sodium chloride, an orange solution from which compound **8** was isolated as an orange solid in 82% yield (Figure 4). The neutral character of **8** was confirmed by conductivity measurements in acetone. In addition, the MALDI-Tof mass spectrum showed a peak at *m*/*z* 441.29, corresponding to the protonated molecular ion. The ^1^H and ^13^C{^1^H} NMR spectra are consistent with the unsymmetrical structure resulting from the conformational constraint imposed by the coordination of the alkoxo fragment to the metal center. It is noticeable the deshielding of the olefinic proton resonance of the 1,5-cyclooctadiene ligand trans to the alkoxy group in the ^1^H NMR spectra, from 4.61 and 4.54 ppm in **7** to 5.11 and 5.05 ppm in **8**, which is diagnostic for the coordination of the alkoxo group to the metal center [44].

The unbridged bis-carbene compound [Ir(cod){MeIm(CH_2_)_3_OH}_2_]Cl (**9**) has been synthesized via deprotonation of the imidazolium salt **D** by the bridging ethoxo ligands in the dimer [Ir(μ-OEt)(cod)]_2_, generated in situ by reaction of [Ir(μ-Cl)(cod)]_2_ with NaH in ethanol, in the presence of an excess of NaOEt (Figure 4). The complex was obtained as an orange-yellow microcrystalline solid with a 72% yield after the removal of the inorganic salts.

The cationic character of the compound was established by conductivity measurements. However, the molar conductivity value, 55 Ω^−1^cm^2^mol^−1^, is appreciably below that expected for 1:1 electrolytes in acetone, which is attributed to the formation of ionic pairs due to the presence of polar wingtips at the NHC ligands. In addition, the mass spectra (ESI^+^, MeOH) showed the molecular ion with the right isotopic pattern at *m*/*z* 581.2468. The ^1^H NMR of **9** exhibited duplicated resonances for both cod and NHC ligands, which evidenced the presence of two symmetrical isomers. For example, the =CH protons of the imidazole-2-ylidene rings showed two sets of resonances in a 70/30 ratio, and two low field resonances at δ 177.8 and 176.6 ppm for the carbene carbon atoms of both isomers were observed in the ^13^C{^1^H} NMR. The presence of two diastereomers for **9** is derived from the relative disposition of the 3-hydroxypropyl functionalized NHC ligands: the *up-down* isomer (*C*_2_ symmetry) and the *up-up* isomer (*C*_s_ symmetry) having antiparallel and parallel arrangement of the carbene ligands, respectively (Figure 3). It should be noted that related compounds [Ir(cod)(NHC∩Z)_2_]^+^ (∩Z = 2-methoxybenzyl; pyridin-2-ylmethyl and quinolin-8-ylmethyl) also exist as two diastereomers that interconvert in solution in which the “up-down” is the major diastereomer [40].

The assignment of the ^1^H and ^13^C{^1^H} resonances of both diastereomers of **9** was achieved by a combination of the ^1^H−^1^H COSY, ^13^C APT, and ^1^H−^13^C HSQC spectra. Remarkably, the bidimensional ^1^H−^1^H-NOESY spectrum showed strong exchange cross-peaks between all types of protons for both species, together with weak NOE cross-peaks, indicating that both diastereomers interconvert in solution (see the Appendix A).

### 2.3. Dehydrogenation of Glycerol to Lactic Acid

The described iridium(I) complexes **1** and **3**–**9** have been evaluated as catalysts for the acceptorless dehydrogenation of glycerol to produce lactic acid and H_2_. The influence of the functional group in the bridge on the catalytic activity of bis-NHC iridium complexes, as well as that of the auxiliary ligands, has been investigated and compared with that of complexes featuring functionalized monodentate NHC ligands. The reactions were performed in an open flask using net glycerol and KOH as a base in the presence of the iridium catalyst. The catalytic performance of the catalysts with different base/catalyst ratios at diverse temperatures is summarized in Table 2.

No formation of lactate (LA) or other glycerol derivatives was observed with catalysts **1** and **4** in the absence of a base (entries 1–2) or in the absence of the catalyst (entry 3) after a prolonged reaction time at 115 °C. On the other hand, blank tests with the imidazolium salts A-D ([Fig molecules-27-07666-ch001]) provide no glycerol conversion. Preliminary catalytic tests with the bis-NHC carbonyl complexes **3** and **5** (1 mol%) using KOH as a base in glycerol/water (1:1) under very mild temperature conditions gave good glycerol conversions albeit moderate selectivities for LA, due to the formation of 1,2-propane diol, with turnover numbers for the production of lactate, TON_LA_, of ca. 200 (entries 4 and 5) which shows the potential of this family of catalysts for acceptorless dehydrogenation of glycerol. In light of these results, the reaction conditions were optimized in order to reduce the catalyst loading and improve the selectivity towards lactate salt. So, to compare the catalytic performance of the iridium(I) catalysts, we have selected the following standard catalytic reaction conditions: 5 mmol of KOH, catalyst loading of 0.2 mol% (relative to the limiting reagent KOH), in neat glycerol at 130 °C.

Under the standard catalytic reaction conditions, all complexes **1** and **3**–**9** are active for the acceptorless dehydrogenation of neat glycerol (entries 6–13) with good selectivities, in the range 86–100%, for LA. In the cases where selectivity does not reach 100%, the only byproduct detected is 1,2-propanediol. The more active catalysts **4**, **5** and **7** provided conversions higher than 60% in only 2 h with selectivities higher than 85%. Among them, catalyst [Ir(CO)_2_{(MeImCH_2_)_2_CHOH}]Br (**5**) exhibited the best catalytic performance with an adequate balance of activity and selectivity with a TON_LA_ superior to 400 (entry 9). In general, carbonyl complexes are more active than diene complexes. As for the bis-NHC complexes, bridge-functionalized complexes are more active than compound **6**, featuring an unfunctionalized bis-NHC ligand. In addition, the hydroxy-functionalized bis-NHC complexes **4** and **5**, are much more active than the carboxylate-functionalized complexes **1** and **3**. The excellent catalytic performance of [IrCl(cod){MeIm(CH_2_)_3_OH}] (**7**), TON_LA_ of 344, contrast with that of the alkoxo compound **8** and the unbridged bis-NHC compound **9** (entries 11–13).

Other inorganic bases, such as NaOH or Cs_2_CO_3_, have also been investigated under standard conditions using catalyst **7**. The catalyst performance was much worse with Cs_2_CO_3_, and with NaOH, it took 6 h to achieve the same conversion (entries 14–15). The better performance of KOH is likely related to the higher solubility of potassium hydroxide in glycerol. Increasing the temperature from 130 to 150 °C resulted in an increase in activity, reaching similar conversions in only 0.75 h and selectivities to LA higher than 90% (entries 16–20). Under these conditions, catalyst **5** was the most efficient, with total conversion and a 93% selectivity to LA (entry 19). On the other hand, the influence of the amount of KOH on the catalytic activity has also been studied using the same amount of catalyst **5**(0.01 mmol, entry 19). A decrease in the amount of KOH leads to a drastic decrease in LA formation (entries 21–22), and the activity did not improve when the amount of KOH was increased to 6 mmol (entry 23). Interestingly, it was possible to reduce the catalyst load to 0.07 mol% achieving good conversion in 24 h and full conversion to LA in 72 h. Excellent selectivity to LA was also attained when the reaction was performed in a glycerol/H_2_O (1:1), albeit with moderate conversion (entries 24–26). The catalyst loading can be reduced to 70 ppm (0.007 mol%), attaining 74% conversion in 72 h with complete selectivity to LA. Further reduction of the catalyst loading to 14 ppm (0.0014 mol%) allows a remarkable TON_LA_ of 15,000 (entries 27 and 28).

The catalytic experiments of Table 2 were performed in an open flask allowing the release of the H_2_(g) produced in the dehydrogenation of glycerol. The catalytic dehydrogenation reactions were also performed in a close microreactor equipped with a pressure transducer to monitor the H_2_(g) evolution along the catalytic reaction. The reactions were carried out in net glycerol using KOH as a base (5 mmol) and a catalyst loading of 0.2 mol% of iridium catalysts at 130 °C. In general, the reactions in a closed reactor are slower than in an open system. However, good conversions were obtained after 5 h. The reaction profiles for catalysts [Ir(cod){(MeIm)_2_CHCOO}] (**1**), [Ir(cod){(MeImCH_2_)_2_CHOH}]Br (**4**) and [IrCl(cod){MeIm(CH_2_)_3_OH}] (**7**) are shown in Figure 4. Catalysts **1** and **7** exhibited a similar behavior which contrasts with the excellent activity of catalyst **4** under these conditions. Nevertheless, the kinetic profiles are quite similar for all of them, showing a very short induction period for catalyst preactivation. In all three cases, the amount of hydrogen released after 5 h of reaction is below the stoichiometric limit determined by the KOH added (5 mmol), which results in conversions of 94% (**4**), 84% (**1**) and 81% (**7**). The ^1^H NMR spectra in D_2_O of the resultant reaction mixture for all three catalytic tests showed the presence of the intermediate 1,2-propanediol in ca. 10%, but not degradation products derived from glycerol overoxidation or glycerol C–C cleavage (see Appendix A).

These results show that in the presence of molecular hydrogen (closed flask), not only the activity decreases but also the selectivity due to the increases in the amount of propanediol, which is consistent with the commonly accepted mechanism for glycerol dehydrogenation by iridium catalysts [20]. We assumed that glycerol conversion likely proceeds through the acceptorless iridium-catalyzed dehydrogenation of glycerol to dihydroxyacetone or glyceraldehyde, which interconverts in the presence of a base. Subsequent base-catalyzed dehydration of glyceraldehyde would lead to pyruvaldehyde which reacts by an intramolecular Cannizzaro reaction to yield LA. However, the hydrogenation of the enol-pyruvaldehyde tautomer by the iridium catalyst would lead to 1,2-propanediol, which is the main byproduct of the reaction and is favored under hydrogen pressure. Subsequently, the dehydrogenation of 1,2-propanediol, which is inhibited under hydrogen, results in the formation of pyruvaldehyde (Figure 5).

## 3. Experimental Section

### 3.1. Materials and Methods

Experiments under an atmosphere of argon were carried out using Schlenk techniques or in a dry box. Solvents were distilled immediately prior to use from the appropriate drying agents or obtained from a Solvent Purification System (Innovative Technologies). Oxygen-free solvents were employed throughout. CDCl_3_, CD_2_Cl_2_ and bencene-*d*_6_ were dried using activated molecular sieves. Methanol-*d*_4_ (<0.02% D_2_O) was purchased from Eurisotop and used as received. The imidazolium salts, 1,1-bis(*N*-methylimidazolium)-acetate-bromide, [(MeImH)_2_CHCOO]Br (**A**) [30], 1,1′-(2-hydroxy-1,3-propanediyl)bis[3-methyl-1H-imidazolium]dibromide, [(MeImHCH_2_)_2_CHOH]Br_2_ (**B**) [39], 1,1′-methylene-bis(3-methylimidizolium)diiodide, [(MeIm)_2_CH_2_]I_2_ (**C**) [35], and 1-(3-hydroxypropyl)-3-methyl-1*H*-imidazol-3-ium chloride, [MeImH(CH_2_)_3_OH]Cl (**D**) [36], were prepared according to the literature. The starting materials [Ir(μ-Cl)(cod)]_2_ [45], and [Ir(μ-OMe)(cod)]_2_ [46], and compound [Ir(cod){(MeIm)_2_CHCOO}] (**1**) [30], were prepared following the procedures previously reported.

### 3.2. Scientific Equipment

C, H, and N analyses were carried out in a PerkinElmer 2400 Series II CHNS/O analyzer. Infrared spectra were recorded on an FT-PerkinElmer Spectrum One spectrophotometer using Nujol mulls between polyethylene sheets. ^1^H NMR spectra were recorded on a Bruker Avance 300 (300.128 MHz and 75.479 MHz) or Bruker Avance 400 (400.130 MHz and 100.613 MHz). NMR chemical shifts are reported in ppm relative to tetramethylsilane and are referenced to partially deuterated solvent resonances. Coupling constants (*J*) are given in hertz. Spectral assignments were achieved by a combination of ^1^H−^1^H COSY, ^13^C APT, ^1^H−^13^C HSQC and ^1^H−^13^C HMBC experiments. High-resolution electrospray ionization mass spectra (HRMS-ESI) were recorded using a Bruker MicroToF-Q equipped with an API-ESI source and a Q-ToF mass analyzer, which leads a maximum error in the measurement of 5 ppm, using sodium formate as reference. MALDI-TOF mass spectra were obtained on a Bruker MICROFLEX spectrometer using DIT, ditranol, 1,8-dihidroxi-9,10-dihydroanthracen-9-one, as matrix [47]. Conductivities were measured in ca. 5 × 10^−4^ M acetone solutions of the complexes using a Philips PW 9501/01 conductimeter.

### 3.3. Compound Synthesis

#### 3.3.1. Synthesis of [Ir(CO)(cod){(MeIm)_2_CHCOO}] (**2**), Figure 5

Carbon monoxide was bubbled through a solution of [Ir(cod){(MeIm)_2_CHCOO}] (**1**) (25.0 mg, 0.048 mmol) in CH_2_Cl_2_ (5 mL) for 10 min to give a pale-yellow solution. The concentration of the solution to ca. 1 mL and the slow addition of diethyl ether afforded the compound as a yellow solid that was filtered, washed with cold diethyl ether (3 × 3 mL) and dried in vacuo. Yield: 18.9 mg, 72%. Anal. Calc. for C_19_H_23_IrN_4_O_3_·CH_3_OH: C, 41.44; H, 4.69; N, 9.66. Found: C, 41.56; H, 4.61; N, 9.70. MS (ESI+, CH_2_Cl_2_/MeOH, *m*/*z*): calcd. for C_19_H_24_IrN_4_O_3_ [M+H]^+^: 549.15, found: 549.10. IR (ATR, cm^−1^): 1944 (CO), 1635 (COO). ^1^H NMR (298 K, 300 MHz, CD_3_OD): δ 7.44 (d, *J* = 2.0, 2H, CH), 7.28 (d, *J* = 2.0, 2H, CH), 6.73 (s, 1H, CHCOO), 3.90 (br, 4H, =CH cod), 3.68 (s, 6H, NCH_3_), 2.64–2.45 (m, 4H, >CH_2_ cod), 2.41–2.21 (m, 4H, >CH_2_ cod). ^13^C{^1^H} NMR (298 K, 75 MHz, CD_3_OD): δ 179.0 (CO), 150.5 (C_NCN_), 124.3, 123.4 (CH), 76.6 (*C*HCOO), 71.3 (=CH cod), 38.4 (NCH_3_), 35.0 (>CH_2_ cod).

**Figure 5 molecules-27-07666-f005:**
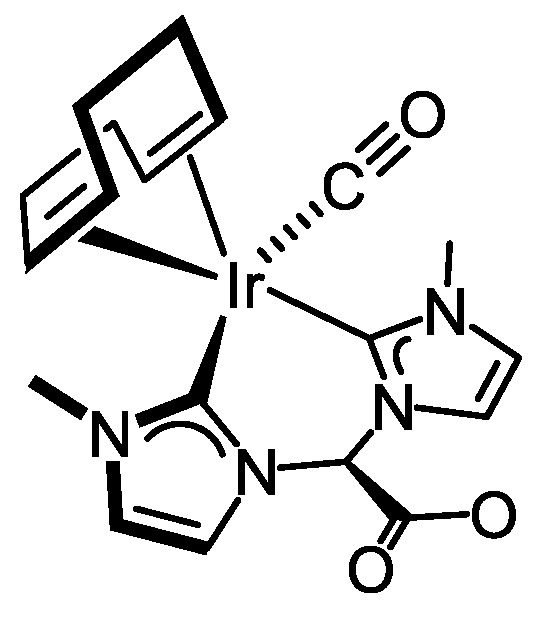
Complex **2**.

#### 3.3.2. Synthesis of [Ir(CO)_2_{(MeIm)_2_CHCOO}] (**3**), Figure 6

A solution of [Ir(cod){(MeIm)_2_CHCOO}] (**1**) (20.0 mg, 0.038 mmol) in CH_2_Cl_2_ (5 mL) was stirred for 16 h under carbon monoxide atmosphere to give a clear yellow solution. The concentration of the solution to ca. 1 mL and slow addition of diethyl ether afforded the compound as a yellow solid that was filtered, washed with cold diethyl ether (3 × 3 mL) and dried in vacuo. Yield: 9.9 mg, 55%. Anal. Calc. for C_12_H_11_IrN_4_O_4_: C, 30.83; H, 2.37; N, 11.99. Found: C, 30.95; H, 2.28; N, 11.89. MS (ESI+, CH_2_Cl_2_/MeOH, *m*/*z*): calcd. for C_11_H_12_IrN_4_O_3_ [M-CO+H]^+^: 441.05, found: 441.32. IR (ATR, cm^−1^): 2056, 1987 (CO), 1647 (COO).^1^H NMR (298 K, 300 MHz, CD_3_OD): δ 7.57 (d, *J* = 1.9, 2H, CH), 7.39 (d, *J* = 1.9, 2H, CH), 6.78 (s, 1H, CHCOO), 3.93 (s, 6H, NCH_3_). ^13^C{^1^H} NMR (298 K, 75 MHz, CD_3_OD): δ 183.2 (CO), 170.2 (C_NCN_), 170.0 (COO), 124.2, 124.0 (CH), 76.3 (*C*HCOO), 36.1 (NCH_3_).

**Figure 6 molecules-27-07666-f006:**
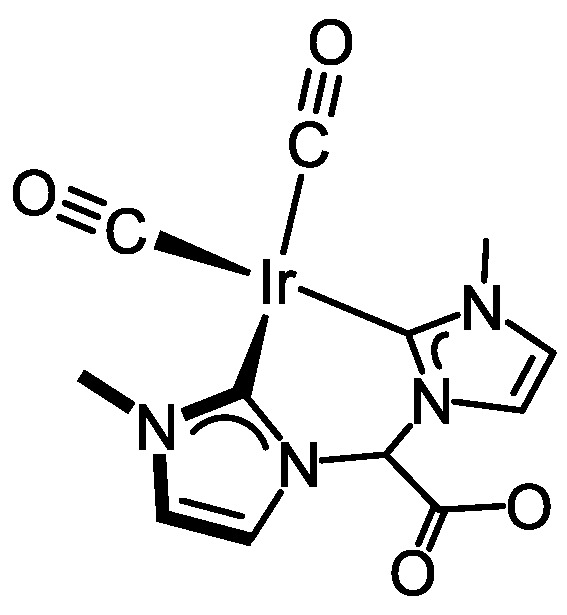
Complex **3**.

#### 3.3.3. Synthesis of [Ir(cod){(MeImCH_2_)_2_CHOH}]Br (**4**), Figure 7

NaH (22.0 mg, 60% oil dispersion, 0.550 mmol) and [Ir(µ-Cl)(cod)]_2_ (84.1 mg, 0.125 mmol) were dissolved independently in ethanol (5 mL). The solutions were combined and stirred for 15 min to give a yellow solution which was transferred via cannula to a solution of [{(MeImH)CH_2_}_2_CHOH]Br_2_ (95.7 mg, 0.250 mmol) in ethanol (5 mL). After stirring for 16 h, the orange solution was brought to dryness under a vacuum, and the orange residue was extracted with dichloromethane (7 mL) and then filtered via cannula. The resulting orange solution was concentrated to ca. 1 mL under reduced pressure. The slow addition of cold diethyl ether resulted in a microcrystalline shiny orange solid, which was washed with diethyl ether (3 × 3 mL) and dried in vacuo. Yield: 120.1 mg, 80%. Anal. Calc. for C_19_H_28_BrIrN_4_O: C, 38.00; H, 4.70; N, 9.33. Found: C, 37.99; H, 4.69; N, 9.33. MS (ESI+, CH_2_Cl_2_/MeOH, *m*/*z*): calcd. for C_19_H_28_IrN_4_O [M]^+^: 521.19, found: 521.11. The compound was obtained as two isomers in a 2/1 ratio. *Major isomer:* ^1^H NMR (298 K, 400 MHz, CD_2_Cl_2_): δ 7.37 (d, *J* = 1.9, 2H, CH), 6.80 (d, *J* = 1.9, 2H, CH), 6.25 (s, 1H, OH), 4.70 (br, 2H, NCH_2_), 4.47 (br, 2H, NCH_2_), 4.47 (s, C*H*OH), 4.09 (br, 4H, =CH cod), 3.80 (s, 6H, NCH_3_), 2.43–2.23 (br, 4H, >CH_2_ cod), 2.16–1.95 (br, 4H, >CH_2_ cod). ^13^C{^1^H} NMR (298 K, 101 MHz, CD_2_Cl_2_): δ 178.6 (C_NCN_), 126.1, 120.9 (CH), 77.2 (=CH cod), 76.0 (=CH cod)*,* 67.1 (*C*HOH), 55.8 (CH_2_) 37.8 (NCH_3_), 31.8, 37.8 (>CH_2_ cod). *Minor isomer:* ^1^H NMR (298 K, 400 MHz, CD_2_Cl_2_): δ 7.23 (d, *J* = 1.8, 2H, CH), 6.88 (d, *J* = 1.9, 2H, CH), 6.25 (s, 1H, OH), 4.75 (br, 2H, NCH_2_), 4.52 (br, 2H, NCH_2_), 4.47 (s, C*H*OH), 4.14 (br, 4H, =CH cod), 3.81 (s, 6H, NCH_3_), 2.43–2.23 (br, 4H, >CH_2_ cod), 2.16–1.96 (br, 4H, >CH_2_ cod). ^13^C{^1^H} NMR (298 K, 101 MHz, CD_2_Cl_2_): δ 178.3 (C_NCN_), 124.0, 122.1 (CH), 77.9 (=CH cod), 76.2 (=CH cod), 67.1 (*C*HOH), 59.1 (CH_2_), 37.8 (NCH_3_), 31.7, 31.7 (>CH_2_ cod).

**Figure 7 molecules-27-07666-f007:**
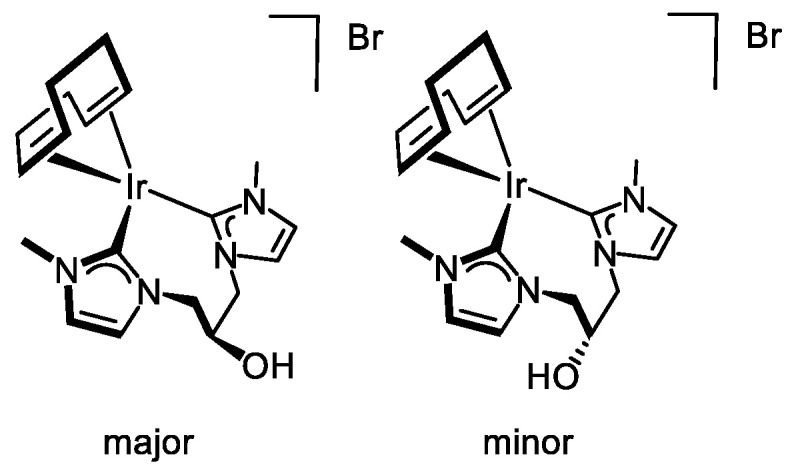
Diastereomers of complex **4**.

#### 3.3.4. Synthesis of [Ir(CO)_2_{(MeImCH_2_)_2_CHOH}]Br (5), Figure 8

Carbon monoxide was bubbled through a solution of complex [Ir(cod){(MeIm_2_CH_2_)_2_CHOH}]Br (**4**) (90.0 mg, 0.164 mmol) in CH_2_Cl_2_ (5 mL) for 10 min to give a pale yellow solution which was concentrated to ca. 1 mL under reduced pressure. The addition of cold diethyl ether (2 mL) afforded a pale-yellow solid, which was filtered, washed with diethyl ether (3 × 3 mL) and dried in vacuo. Yield: 72.3 mg, 80%. Anal. Calc. for C_13_H_16_BrIrN_4_O_3_: C, 28.47; H, 2.94; N, 10.22. Found: C, 28.39; H, 2.68; N, 10.18. MS (ESI+, CH_2_Cl_2_/MeOH, *m*/*z*): calcd. for C_13_H_16_IrN_4_O_3_ [M]^+^: 469.08, found: 469.00. IR (ATR, cm^−1^): 2058, 1985 (CO). The compound was obtained as two isomers in a 4/1 ratio. *Major isomer:* ^1^H NMR (298 K, 300 MHz, CD_2_Cl_2_): δ 7.54 (d, *J* = 1.9, 2H, CH), 6.93 (d, *J* = 1.8, 2H, CH), 6.22 (s, 1H, OH), 4.69–4.50 (br, 4H, CH_2_), 4.41 (s, 1H, C*H*OH), 3.79 (s, 3H, NCH_3_). ^13^C{^1^H} NMR (298 K, 75 MHz, CD_2_Cl_2_): δ 182.3 (CO), 169.3 (C_NCN_), 127.6, 122.5 (CH), 65.3 (CHOH), 56.4 (CH_2_), 38.7 (NCH_3_). *Minor isomer:* ^1^H NMR (298 K, 300 MHz, CD_2_Cl_2_): δ 7.46 (d, *J* = 1.8, 2H, CH), 7.02 (d, *J* = 1.8, 2H, CH), 6.22 (s, 1H, OH), 4.69–4.50 (br, 4H, CH_2_), 4.41 (s, 1H, C*H*OH), 3.80 (s, 6H, NCH_3_). ^13^C{^1^H} NMR (298 K, 75 MHz, CD_2_Cl_2_): δ 182.3 (CO), 169.3 (C_NCN_), 125.4, 123.8 (CH), 70.5 (CHOH), 59.4 (CH_2_), 38.7 (NCH_3_).

**Figure 8 molecules-27-07666-f008:**
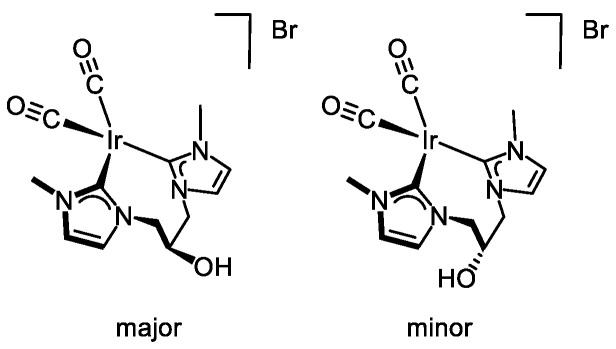
Diastereomers of complex **5**.

#### 3.3.5. Synthesis of [Ir(cod){(MeIm)_2_CH_2_}]I (**6**), Figure 9

To a suspension of [Ir(μ-Cl)(cod)]_2_ (73.2 mg, 0.11 mmol) in ethanol (5 mL) was slowly added a solution of NaH (13 mg, 0.48 mmol) in ethanol (2 mL), and the mixture was stirred at room temperature for 30 min. Then, the imidazolium salt [(MeImH)_2_CH_2_]I_2_ (94.2 mg, 0.218 mmol) was added and the suspension was stirred for 24 h. The solvent was pumped off and the residue was extracted with CH_2_Cl_2_ (2 × 5 mL). The solution was concentrated under a vacuum to 1 mL, and then diethylether (10 mL) was added to give the compounds as orange-yellow solids that were washed with diethylether (2 × 5 mL) and dried under vacuum. Yield: 83.2 mg, 63%. Anal. Calc. for C_17_H_24_IIrN_4_: C 33.83; H, 4.01; N, 9.28. Found: C, 33.92; H, 4.40; N, 9.01. MS (ESI+, MeOH, *m*/*z*): calcd. for C_17_H_24_IrN_4_ [M]^+^: 477.17, found: 477.20. ^1^H NMR (298 K, 400 MHz, CD_2_Cl_2_): δ 7.26 (d, *J*_H-H_ = 2.0, 2H, CH Im), 6.81 (d, *J*_H-H_ = 2.0, 2H, CH), 6.03 (d, *J*_H-H_ = 11.9, 1H, NCH_2_N), 5.82 (d, *J*_H-H_ = 11.9, 1H, NCH_2_N), 3.82 (m, 4H, =CH cod), 3.79 (s, 6H, NCH_3_), 2.32–2.06 (m, 4H, >CH_2_ cod), 2.01–1.80 (m, 4H, >CH_2_ cod). ^13^C{^1^H} NMR (298 K, 101 MHz, CD_2_Cl_2_): δ 172.1 (C_NCN_), 122.2, 119.2 (CH Im), 63.9 (=CH cod) and (NCH_2_N), 39.3 (NCH_3_), 33.2 (>CH_2_ cod).

**Figure 9 molecules-27-07666-f009:**
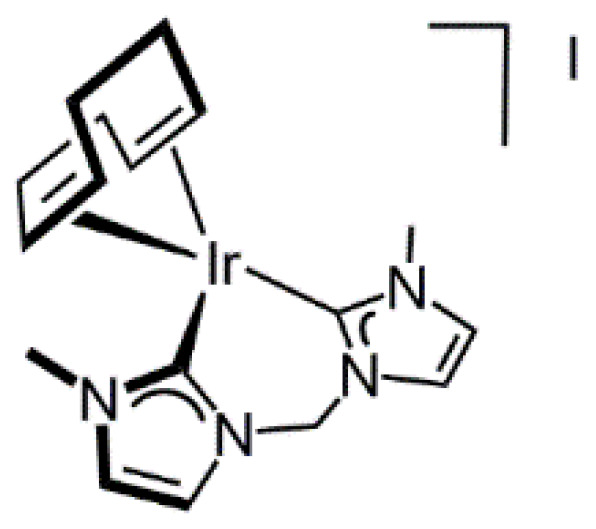
Complex **6**.

#### 3.3.6. Synthesis of [IrCl(cod){MeIm(CH_2_)_3_OH}] (**7**), Figure 10

A mixture of [{Ir(μ-OMe)(cod)}_2_] (189 mg, 0.28 mmol) and [MeImH(CH_2_)_3_OH]Cl (100 mg, 0.57 mmol) in THF (10 mL) was stirred overnight at room temperature to give an orange suspension. The solid was removed by filtration, and the resulting orange solution was evaporated to dryness. Treatment of the yellow residue with pentane rendered a yellowish solid, which was separated by decantation, washed with pentane, and dried in a vacuum. Yield: 233 mg, 86%. Anal. Calcd for C_15_H_24_ClN_2_OIr: C, 37.85; H, 5.08; N, 5.89. Found: C, 37.68; H, 5.03; N, 5.86. MS (MALDI-Tof, DIT matrix, CH_2_Cl_2_, *m*/*z*): calcd. for C_15_H_25_N_2_OIr [M-Cl+H]^+^: 441.59, found: 441.29, Λ_M_ (acetone): 12 Ω^−1^ cm^2^ mol^−1^. ^1^H NMR (298 K, 300 MHz, CDCl_3_): *δ* 6.85, (q, *J* = 1.9, 2H, CH Im), 5.22 (dd, *J* = 11.5, 4.1, 1H, NCH_2_), 4.61, 4.54 (m, 2H, =CH cod), 4.01 (dt, *J* = 13.8, 4.1, 1H, NCH_2_), 3.94 (s, 3H, MeIm), 3.56, 3.47 (m, 2H, CH_2_O), 3.14 (m, 1H, OH), 2.98, 2.90 (m, 2H, =CH cod), 2.22 (m, 4H, >CH_2_ cod), 2.11–1.88 (m, 2H, CH_2_), 1.80–1.54 (m, 4H, >CH_2_ cod). ^13^C{^1^H} NMR (298 K, 75 MHz, CDCl_3_): *δ* 180.2 (NCN), 122.5, 119.3 (CH Im), 85.1, 84.4 (=CH cod), 56.7 (CH_2_O), 52.5, 52.2 (=CH cod), 46.1 (NCH_2_), 37.6 (MeIm), 34.0, 33.3 (>CH_2_ cod), 32.7 (CH_2_), 30.0, 29.4 (>CH_2_ cod). The molecular structure of **7** has been reported in ref. [48].

**Figure 10 molecules-27-07666-f010:**
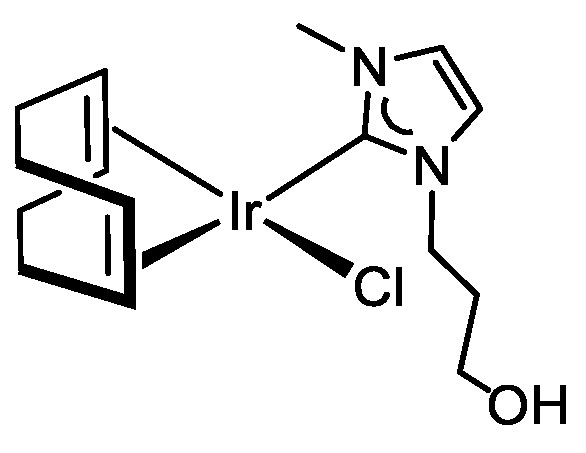
Complex **7**.

#### 3.3.7. Synthesis of [Ir(cod){κ^2^C,O-{MeIm(CH_2_)_3_O}] (**8**), Figure 11

NaH (5.3 mg, 0.21 mmol) was added to a solution of [IrCl(cod){MeIm(CH_2_)_3_OH}] (**7**) (100 mg, 0.21 mmol) in THF (10 mL). The mixture was stirred for 4 h at room temperature. After the elimination of the inorganic salts by filtration, the resulting yellow solution was concentrated until 1 mL. Slow addition of n-hexane (3 mL) afforded an orange solid, which was separated by decantation, washed with n-hexane, and dried in a vacuum. Yield: 76 mg, 82%. Anal. Calcd for C_15_H_23_N_2_OIr: C, 40.99; H, 5.27; N, 6.37. Found: C, 40.86; H, 5.31; N, 6.39. MS (MALDI-Tof, DIT matrix, CH_2_Cl_2_, *m*/*z*): calcd. for C_15_H_25_N_2_OIr [M+2H]^+^: 441.59, found: 441.29; calcd. for C_15_H_27_N_2_O_2_Ir [M+2H+H_2_O]^+^: 459.60, found: 459.21. Λ_M_ (acetone): 9 Ω^−1^ cm^2^ mol^−1^.^1^H NMR (298 K, 300 MHz, C_6_D_6_): *δ* 6.23, 6.10 (s, 2H, CH Im), 5.24 (m, 1H, NCH_2_), 5.11, 5.05 (m, 2H, =CH cod), 3.65 (m, 3H, NCH_2_ y CH_2_O), 3.50 (s, 3H, MeIm), 2.99 (m, 2H, =CH cod), 2.29 (m, 4H, >CH_2_ cod), 1.93, 1.80 (m, 2H, >CH_2_), 1.76–1.56 (m, 4H, >CH_2_ cod). ^13^C{^1^H} NMR (298 K, 75 MHz, C_6_D_6_): *δ* 180.7 (NCN), 121.9, 119.3 (CH Im), 84.9, 84.2 (=CH cod), 56.9 (CH_2_O), 51.6, 51.2 (=CH cod), 46.4 (NCH_2_), 37.0 (MeIm), 34.4 (CH_2_), 33.6, 33.2, 30.4, 29.6 (>CH_2_ cod). 

**Figure 11 molecules-27-07666-f011:**
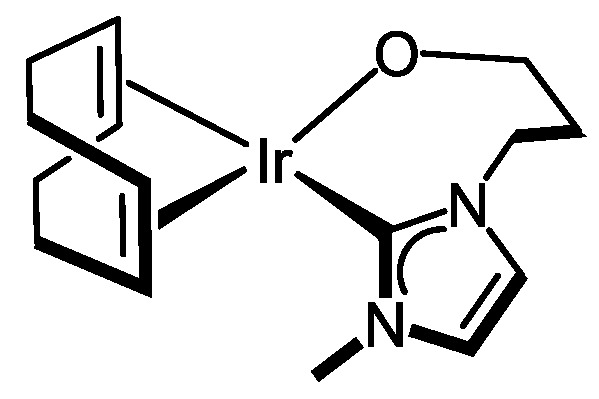
Complex **8**.

#### 3.3.8. Synthesis of [Ir(cod){MeIm(CH_2_)_3_OH}_2_]Cl (**9**), Figure 12

A solution of NaH (24 mg, 1.0 mmol) in ethanol (2 mL) was slowly added to a suspension of [Ir(μ-Cl)(cod)]_2_ (101 mg, 0.150 mmol) in ethanol (5 mL), and the mixture was stirred at room temperature for 30 min. Then, the imidazolium salt [MeImH(CH_2_)_3_OH]Cl (106 mg, 0.6 mmol) was added, and the suspension was stirred for 48 h. The solvent was pumped off, and the residue was extracted with CH_2_Cl_2_ (2 × 5 mL). The solution was concentrated under a vacuum to 1 mL, and then diethyl ether (10 mL) was added to give an orange-yellow solid, which was separated by decantation, washed with diethyl ether (2 × 5 mL) and dried under vacuum. Yield: 140 mg, 72%. HRMS (ESI+, MeOH, *m*/*z*): calcd. for C_22_H_36_IrN_4_O_2_ [M]^+^: 580.2507, found: 581.2468. Λ_M_ (acetone): 54 Ω^−1^ cm^2^ mol^−1^. The compound was obtained as two isomers in an 80/20 ratio. *Major isomer:* ^1^H NMR (298 K, 300 MHz, CD_2_Cl_2_): *δ*: 7.05 (d, *J* = 1.0, 1H, CH Im), 6.94 (d, *J* = 1.0, 1H, CH Im), 5.43 (broad, 1H, OH), 4.75, (ddd, *J* = 12.4, 11.7, 5.2, 1H, NCH_2_), 4.25 (ddd, *J* = 12.9, 11.7, 5.2, 1H, NCH_2_), 3.91 (m, 2H, =CH cod), 3.89 (s, 3H, MeIm), 3.91, 3.78 (m, 2H, =CH cod), 3.76, 3.64 (m, 2H, *CH_2_*OH), 2.46, 1.86 (m, 2H, NCH_2_*CH_2_*), 2.23, 1.95 (m, 4H, >CH_2_ cod), 2.04, 1.96 (m, 4H, >CH_2_ cod). ^13^C{1H} NMR (298K, 75 MHz, CD_2_Cl_2_): *δ* 177.8 (NCN), 123.1, 121.2 (CH Im), 77.3, 76.1 (=CH cod), 58.8 (CH_2_OH), 49.6 (NCH_2_), 38.3 (CH_3_), 34.7 (NCH_2_*CH_2_*), 32.0, 31.6 (>CH_2_ cod). *Minor isomer:* ^1^H NMR (298 K, 300 MHz, CD_2_Cl_2_): *δ*: 7.19 (m, 1H, CH Im), 7.01 (m, 1H, CH Im), 4.41, 4.17 (m, 2H, NCH_2_), 4.10, 3.57 (m, 2H, =CH cod), 4.07 (s, 3H, MeIm), 4.10, 3.57 (m, 4H, = CH cod), 2.6–1.8 (m, 4H, >CH_2_ cod). ^13^C{^1^H} NMR (298K, 75 MHz, CD_2_Cl_2_): *δ* 176.6 (NCN), 123.4, 121.8 (CH Im), 78.2, 74.5 (=CH cod), 48.6 (NCH_2_), 38.9 (CH_3_). 

**Figure 12 molecules-27-07666-f012:**
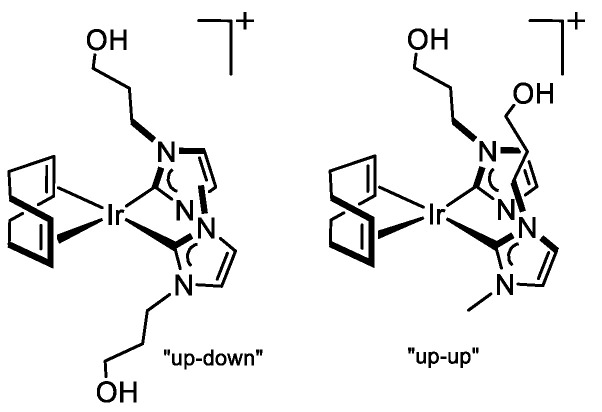
Diastereomers of complex **9**.

### 3.4. General Procedure for the Acceptorless Dehydrogenation of Glycerol

The catalytic reactions were carried out in 25 mL round bottom flasks fitted with a reflux condenser open to the atmosphere. In a typical experiment, the flask was charged under an argon atmosphere with glycerol (1 mL, 13.7 mmol), KOH (312 mg, 5 mmol) and the iridium catalyst and then placed in a thermostatic bath at the desired temperature. The reaction mixture was stirred for the required reaction time and then cooled to room temperature. The resulting suspension was dissolved in 4 mL of MeOH, and then a 0.2 mL aliquot was transferred to a flask. The solvent was removed under vacuum, and the residue was dissolved in D_2_O (0.5 mL), and then NaOAc^.^3H_2_O (30 mg, 0.22 mmol) was added as an internal standard. The yield, with respect to the maximum number of mmol of glycerol that can be transformed, is determined by the amount of base (5 mmol), and the selectivity (lactate salt and 1,2-propanediol) was determined by ^1^H NMR.

H_2_(g) evolution was monitored in a close microreactor equipped with a pressure transducer (Man on the Moon series X102 kit microreactor, www.manonthemoontech.com accessed on 7 November 2022). In a typical experiment, a 43 mL microreactor was charged with glycerol (1 mL, 13.7 mmol) under argon and was isothermal stabilized to 130 °C. Then, KOH (312 mg, 5 mmol) and the iridium catalyst (0.01 mmol) were added. Hydrogen evolution was measured until constant pressure. The amount of H_2_(g) (mmol) produced was calculated by using the Ideal Gas Law.

### 3.5. Crystal Structure Determination

Single crystals of **2**, **4**, and **5** for the X-ray diffraction studies were grown by slow diffusion of diethyl ether into a saturated solution of **2** in methanol or slow diffusion of diethyl ether into a concentrated solution of **4** or **5** in dichloromethane. X-ray diffraction data were collected at 100(2) K on a Bruker SMART APEX CCD diffractometer with graphite-monochromated Mo–Kα radiation (λ = 0.71073 Å) using ω rotations. Intensities were integrated and corrected for absorption effects with SAINT–PLUS [49] and SADABS [50] programs, both included in the APEX2 package. The structures were solved by the Patterson method with SHELXS-97 [51] and refined by full-matrix least-squares on F2 with SHELXL-2014 [52] under WinGX [53]. Pitch and yaw angles have been calculated according to the literature [54]. Despite the fact that the poor quality of the single crystals of **4** and their weak diffraction patterns (*θ*_max_ ≈ 23°) lead to a structural determination featuring a low accuracy, the quality of the refined model is good enough to confirm the molecular structure proposed for **4**. On this basis, crystal data and structure refinement are reported in the following, and a brief discussion of the molecular structure of **4** is also provided. CCDC 2206613 (**2**), 2206611 (**4**) and 2206612 (**5**) contain the supplementary crystallographic data for this paper. These data can be obtained free of charge via http://www.ccdc.cam.ac.uk/conts/retrieving.html, accessed on 7th November 2022 (or from the CCDC, 12 Union Road, Cambridge CB2 1EZ, UK; Fax: +44 1223 336033; E-mail: deposit@ccdc.cam.ac.uk).

#### 3.5.1. Crystal Data and Structure Refinement for **2**

C_19_H_23_IrN_4_O_3_·2CH_3_OH, 611.70 g·mol^–1^, orthorhombic, *Pbca*, *a* = 13.0008(7) Å, *b* = 21.0903(12) Å, *c* = 32.4548(18) Å, *V* = 8898.8(9) Å^3^, *Z* = 16, *D*_calc_ = 1.826 g·cm^–3^, *μ* = 6.042 mm^–1^, *F*(000) = 4832, orange prism, 0.200 × 0.095 × 0.065 mm^3^, *θ*_min_/*θ*_max_ 1.255/26.372°, index ranges –16 ≤ *h* ≤ 16, –26 ≤ *k* ≤ 26, –40 ≤ *l* ≤ 40, reflections collected/independent 96,702/9102 [R(int) = 0.0764], *T*_max_/*T*_min_ 0.5056/0.3386, data/restraints/parameters 9102/6/564, GooF(F^2^) 1.197, *R*_1_ = 0.0399 [I>2σ(I)], *wR*_2_ = 0.0704 (all data), largest diff. peak/hole 1.291/–1.241 e·Å^–3^. NHC@C1: pitch, θ 0.2°, yaw, ψ 2.3°; NHC@C9: pitch, θ 0.3°; yaw, ψ 3.7°.

#### 3.5.2. Crystal Data and Structure Refinement for **4**

C_19_H_28_IrN_4_O)_2_Br_2_·3.5CH_2_Cl_2_, 1498.37 g·mol^–1^, monoclinic, *C*2/*c*, *a* = 31.847(2) Å, *b* = 11.7279(9) Å, *c* = 30.685(2) Å, *β* = 113.4620(10)°, *V* = 10,513.2(14) Å^3^, *Z* = 8, *D*_calc_ = 1.893 g cm^–3^, m = 6.977 mm^–1^, *F*(000) = 5816, 0.130 × 0.110 × 0.090 mm^3^, orange prism, *θ*_min_/*θ*_max_1.559/23.318°, index ranges –35 ≤ *h* ≤ 35, –13 ≤ *k* ≤ 13, –34 ≤ *l* ≤ 34, reflections collected/independent 44,589, 7574 [R(int) = 0.0900], *T*_max_/*T*_min_ 0.8727/0.4738, data/restraints/parameters 7574/7/579, GooF(F^2^) 1.124, *R*_1_ = 0.0566 [I>2σ(I)], *wR*_2_ = 0.1109 (all data), largest diff. peak/hole 1.574/–1.126 e·Å^–3^. NHC@C1: pitch 6.6°, yaw 2.0°; NHC@C11: pitch 5.1°, yaw 2.4°.

#### 3.5.3. Crystal Data and Structure Refinement for **5**

C_13_H_16_BrIrN_4_O_3_·CH_2_Cl_2_, 633.33 g·mol^–1^, Monoclinic, *P*2_1_/*c*, *a* = 27.021(9) Å, *b* = 13.060(4) Å, *c* = 11.002(4) Å, *β* = 90.569(4)°, *V* = 3882(2) Å^3^, *Z* = 8, *D*_calc_ = 2.167 g·cm^3^, *μ* = 9.232 mm^–1^, *F*(000) = 2400, yellow prism, 0.180 × 0.108 × 0.070 mm^3^, *θ*_min_/*θ*_max_ 1.507/26.372°, index ranges –33 ≤ *h* ≤ 33, –16 ≤ *k* ≤ 16, –13 ≤ *l* ≤ 13, reflections collected/independent 43,047/7929 [R(int) = 0.0717], *T*_max_/*T*_min_ 0.524/0.329, data/restraints/parameters 7929/0/458, GooF(F^2^) 1.058, *R*_1_ = 0.0537 [I>2σ(I)], *wR*_2_ = 0.1213, largest diff. peak/hole 3.260/–1.684 e·Å^–3^. NHC@C1: pitch 10.2°, yaw 3.2°; NHC@C11: pitch 7.0°, yaw 0.8°.

## 4. Conclusions

A series of mononuclear iridium(I) catalysts featuring carboxylate and hydroxy bridge-functionalized bis-NHC ligands and a cationic unbridged bis-NHC iridium(I) complex bearing 3-hydroxypropyl functionalized NHC ligands have been synthesized and characterized. These catalysts have been shown to be highly active for the selective oxidation of glycerol to lactic acid via acceptorless dehydrogenation of neat glycerol in the presence of KOH as a base. The acetate and hydroxy functions in bis-NHC iridium(I) catalysts increase their solubility in the reaction medium, which results in a significant activity improvement compared to the related iridium(I) catalysts featuring a bridge-unfunctionalized bis-NHC ligand. The hydroxy-functionalized bis-NHC catalysts were found to be much more active than the carboxylate-functionalized ones, and the unbridged bis-NHC iridium(I) catalyst with hydroxyalkyl functionalized NHC ligands. Interestingly, the related neutral chlorido-complex featuring only a hydroxyalkyl-functionalized NHC ligand exhibited a notable catalytic activity. In general, carbonyl complexes are more active than the related diene complexes, with [Ir(CO)_2_{(MeImCH_2_)_2_CHOH}]Br as the most active catalyst affording TONs to LA up to 15,000 at very low catalyst loadings. The high catalytic activity of bridge-functionalized bis-NHC iridium(I) complexes highlights their potential for the development of practical technologies for the transformation of glycerol into value-added chemicals.

## Data Availability

Not available.

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
