# Peer review of "Selective Oxidation of Glycerol via Acceptorless Dehydrogenation Driven by Ir(I)-NHC Catalysts"

_molecules, 2022, doi:10.3390/molecules27227666_

Round 1

Reviewer 1 Report

The article is focused on synthesis of a series of novel Iridium (I)-based complex compounds as catalysts for glycerol conversion into lactate via acceptorless dehydrogenation. On my opinion, the article does not have any significant drawbacks and can be published as it is. It is only necessary to correct the term "pressure transductor" to "pressure transducer" throughout the text.

Author Response

Thank you very much for your comments. The term" pressure transductor" has been replaced by "pressure transducer" throughout the text (pages 12 and 18).

Reviewer 2 Report

The authors report the synthesis and characterization of a series of mononuclear iridium(I)-bis(NHC) complexes featuring functionalized bis-NHC ligands. The new compounds have been fully characterized including X-ray diffraction studies of some of them. Moreover, their catalytic performance as catalyst precursors for the selective oxidation of glycerol to lactic acid (LA) via acceptor less dehydrogenation of neat glycerol in the presence of KOH as base has been studied. The species with the hydroxy-functionalized bis-NHC ligand were found to be much more active than those with the carboxylate-functionalized ones and the unbridged bis-NHC.

In my opinion, this work meets the criteria of quality and interest to be published in Molecules. I only have minor suggestions.

1.- In figure 2, the structures of 4 and 5 are represented in a different scale. That creates some confusion. Perhaps it would be more appropriate to represent them on the same scale.

2.- In table 2, in many cases the selectivity does not reach 100%. What species are formed in addition to LA? Later it is explained that when the experiments are carried out in a closed system  1,2-propanediol is formed as a secondary product. Is that compound also formed in the experiments carried out in an open flask? If yes, it should be mentioned.

Very minor

Page 2, in reference 19, there is an extra space inside the square brackets.

Author Response

Thank you very much for your comments, and a point-by-point response to them is as follows:

1.- In figure 2, the structures of 4 and 5 are represented in a different scale. That creates some confusion. Perhaps it would be more appropriate to represent them on the same scale.

Note that the structures of 4 and 5 are already represented in the same scale. The perceived mismatch is the consequence of the different thermal parameters of atoms in 4 and 5, which makes ellipsoids of 4 look bigger than those of 5.

2.- In table 2, in many cases the selectivity does not reach 100%. What species are formed in addition to LA? Later it is explained that when the experiments are carried out in a closed system 1,2-propanediol is formed as a secondary product. Is that compound also formed in the experiments carried out in an open flask? If yes, it should be mentioned.

1,2-propanediol is the only by-product in the selective oxidation of glycerol to LA via acceptorless dehydrogenation of neat glycerol in the presence of KOH as base. It has also been detected in an open system. This piece of information has now been included in the manuscript on page 11, as follows: “In the cases where selectivity does not reach the 100 %, the only by-product detected is 1,2-propanediol”.

Page 2, in reference 19, there is an extra space inside the square brackets.

Sorry, but there is no space.

Reviewer 3 Report

The manuscript number molecules-2019978 deals with selective oxidation of glycerol via acceptorless dehydrogena-2 tion driven by Ir(I)-NHC catalysts. The paper is well written and readable. The authors present their results in a scientific way. I believe that after some corrections the paper can be published.

Authors need to examine the catalytic properties of the imidazolium salts (A - D), which should be used as a blank test during catalysis investigation. If this has already been done before, add the necessary references.

Author Response

Thank you very much for your comments, a point-by-point response is as follows:

1.- Authors need to examine the catalytic properties of the imidazolium salts (A - D),

As suggested by the reviewer, we have now performed catalytic tests using the imidazolium salts A-D under the same conditions and no catalytic activity was found.

The following sentence has been included in the text (page 10): “On the other hand, blank tests with the imidazolium salts A-D (Chart 1) provide no glycerol conversion”

Round 2

Reviewer 3 Report

The authors corrected manuscript according to reviwers suggestions.